# *In Vitro* Anticancer Properties of Novel Bis-Triazoles

Maysaa M. Saleh [1,2,*], Duaa A. Abuarqoub [3,4], Alaa M. Hammad [5], Md Shahadat Hossan [2,6], Najneen Ahmed [2,7], Nazneen Aslam [4], Abdallah Y. Naser [1], Christopher J. Moody [8], Charles A. Laughton [2] and Tracey D. Bradshaw [2,*]

[1] Department of Applied Pharmaceutical Sciences and Clinical Pharmacy, Faculty of Pharmacy, Isra University, Amman 11622, Jordan
[2] Biodiscovery Institute, School of Pharmacy, University of Nottingham, University Park, Nottingham NG7 2RD, UK
[3] Department of Pharmacology and Biomedical Sciences, Faculty of Pharmacy and Medical Sciences, University of Petra, Amman 11196, Jordan
[4] Cell Therapy Center, The University of Jordan, Amman 11942, Jordan
[5] Department of Pharmacy, Faculty of Pharmacy, Al-Zaytoonah University of Jordan, Amman 11733, Jordan
[6] Department of Medicine, Center for Human Genomics and Precision Medicine, University of Wisconsin—Madison, Madison, WI 53705, USA
[7] Department of Pharmacy, Faculty of Sciences and Engineering, East West University, Dhaka 1212, Bangladesh
[8] School of Chemistry, University of Nottingham, University Park, Nottingham NG7 2RD, UK
* Correspondence: maysaa.saleh@iu.edu.jo or maysaa.saleh@nottingham.ac.uk (M.M.S.); tracey.bradshaw@nottingham.ac.uk (T.D.B.)

**Abstract:** Here, we describe the anticancer activity of our novel bis-triazoles **MS47** and **MS49**, developed previously as G-quadruplex stabilizers, focusing specifically upon the human melanoma MDA-MB-435 cell line. At the National Cancer Institute (NCI), USA, bis-triazole **MS47** (NCS 778438) was evaluated against a panel of sixty human cancer cell lines, and showed selective, distinct multi-log differential patterns of activity, with $GI_{50}$ and $LC_{50}$ values in the sub-micromolar range against human cancer cells. **MS47** showed highly selective cytotoxicity towards human melanoma, ovarian, CNS and colon cancer cell lines; in contrast, the leukemia cell lines interestingly showed resistance to **MS47** cytotoxic activity. Further studies revealed the potent cell growth inhibiting properties of **MS47** and **MS49** against the human melanoma MDA-MB-435 cell line, as verified by MTT assays; both ligands were more potent against cancer cells than MRC-5 fetal lung fibroblasts (SI > 9). Melanoma colony formation was significantly suppressed by **MS47** and **MS49**, and time- and dose-dependent apoptosis induction was also observed. Furthermore, **MS47** significantly arrested melanoma cells at the G0/G1 cell cycle phase. While the expression levels of Hsp90 protein in melanoma cells were significantly decreased by **MS49**, corroborating its binding to the G4-DNA promoter of the Hsp90 gene. Both ligands failed to induce senescence in the human melanoma cells after 72 h of treatment, corroborating their weak stabilization of the telomeric G4-DNA.

**Keywords:** apoptosis assay; bis-triazoles; cell cycle analysis; G4-DNA; Hsp90; melanoma; NCI 60 cell line panel; senescence

## 1. Introduction

Cancer is a group of diseases defined by the uncontrolled multiplication and metastases of malignant cells. If metastases are not contained, they can cause fatalities. The American Cancer Society estimated that >1.9 million new cancer cases would be diagnosed in the United States in 2021, with 608,570 of these patients predicted to die of cancer [1]. It is anticipated that there will be 20.3 million new cases of cancer and 13.2 million cancer-related deaths, by 2030 [2]. Due to the intricacy of cancer etiology, both molecularly targeted and cytotoxic chemotherapeutic drugs are employed to treat cancer [3,4]. However, unpleasant side effects and drug resistance impede many cancer therapies, motivating researchers

to explore revolutionary anti-cancer techniques that target and defeat this pervasive and frequently fatal disease [3,5]. DNA G-quadruplexes' stabilization is among the anti-cancer methods that has the ability to treat all types of cancer. G-quadruplexes (also referred to as G4-DNA) are four-stranded structures formed by guanine-rich nucleic acid sequences, such as the telomeric DNA G-strand and those within the promoter sequences of genes involved in cellular proliferation and survival, such as Hsp90, and oncogenes, such as *c-myc* and *c-kit* [6–8]. G4-DNAs have gained attention as prospective cancer therapeutic targets. It has been hypothesized that ligands that stabilize and bind selectively to the telomeric G4-DNA in cancer cells could limit the telomere elongation process, resulting in senescence (growth arrest) or delayed cell death, as well as the elimination of tumorigenicity [9]. While stabilization of the promoter G-quadruplex of the Hsp90 gene by G4 binders could theoretically downregulate the gene expression and its product Hsp90 chaperone protein, resulting in the deactivation and eventual degradation of its clients known to be involved in oncogenesis, such as kinases and transcription factors, this is not the case in practice [10–12]. It has been proven that the Hsp90 ATPase activity in tumors is stronger and more complex, whereas it remains dormant in normal cells. This leads in the maintenance of homeostasis as well as growth and survival [13,14]. Additionally, this protein is essential for the majority of oncoproteins involved in cancer cell survival and growth, such as serine/threonine kinase (Akt), expressed protein from v-raf-1, murine leukemia viral oncogene homolog 1 (RAF-1), human growth factor-1, apoptosis induction, and other activities in cancer cells, such as angiogenesis and cell cycle inhibition [15]. Previous research has demonstrated that Hsp90 is essential for the development of solid tumors due to its role in promoting malignant transformation. Importantly, the ability of malignant cells to proliferate is largely dependent on the activity of Hsp90 in cancer cells, which has been shown to prevent tumor cells from succumbing to apoptosis [16]. Thus, inhibiting and/or antagonizing Hsp90 protein is a recognized anticancer therapeutic target.

Prior to this, we developed the novel bis-triazoles **MS47** and **MS49** (Figure 1A) as selective G-quadruplex stabilizers (G4 ligands) and antitumor agents [17]. At a concentration of 5.0 M, **MS47** and **MS49** elicit a considerable stabilization of the G-quadruplex produced by the telomeric G-strand and Hsp90a promoter sequence, according to the FRET experiment [18]. In contrast to the well-known G-quadruplex binder RHPS4 [19] (Figure 1B), **MS47** and **MS49** are less successful in stabilizing G-quadruplexes and exhibit poor and moderate binding affinity to both telomeric and Hsp90a promoter sequence G-quadruplexes [17].

In MTT assays [20], potent antiproliferative activities are demonstrated by **MS47** and **MS49** ($GI_{50}$ values $\leq 0.30$ μM and $\leq 80$ nM, respectively) against human cancer colon HCT-116 and pancreatic MiaPaca-2 cell lines, demonstrating more potency than the control RHPS4 ($GI_{50} \leq 5.7$ μM). In addition, a good selectivity for human carcinoma cell lines over the human normal lung fibroblast MRC-5 cell line was demonstrated by **MS47** ($GI_{50} = 2.2$ μM) and **MS49** ($GI_{50} = 1.3$ μM), with selectivity index (SI) scores $\geq 7.3$ and $\geq 16.3$, respectively. While RHPS4 showed a higher $GI_{50}$ value (9.7 μM) against the normal lung MRC-5 cell line in comparison to its $GI_{50}$ values against colon HCT-116 and pancreatic MiaPaca-2 cancer cell lines, with SI scores of 1.7 and 1.8, respectively. Therefore, the selective growth inhibitory activities of **MS47** and **MS49** for human cancer cells over human normal cells are 4.3- and 9.6-fold, respectively, more than the selectivity of the control RHPS4 [17]. This strongly suggests that in addition to stabilizing G-quadruplexes, **MS47** and **MS49** disrupt other molecular targets and mechanisms of action that contribute to anticancer activity. The potent and selective growth inhibitory activities of both ligands make them attractive and promising anticancer medicines in development for cancer treatment.

**A**

**MS47**: NR₂ = Diethylamino-
**MS49**: NR₂ = Piperidinyl-

**B**

**RHPS4**

**Figure 1.** (**A**) Bis-triazoles **MS47** and **MS49** [17]; (**B**) the G-quadruplex binder reported in the literature: RHPS4 [19].

Melanoma in the EU-28 approaches 90,000 new cases annually and is considered to be one of the cancers with the highest rate of increase, with the Scandinavian nations, Switzerland, and Austria serving as epicenters [21,22]. In the coming decades, the number of melanoma patients will stay high due to an aging population and climate change. UV radiation exposure is a known risk factor for this disease [23,24]. Despite the development of medicines (e.g., Vemurafenib) that target the V600E mutation that is responsible for 60% of metastatic melanoma, clinical resistance evolves rapidly and metastatic melanoma remains an intractable unmet clinical need [25]. Consequently, the development of novel therapeutics for melanoma continues to be a top research goal.

In this report, we describe the antitumor activity exhibited by **MS47** against a panel of sixty human tumor cell lines compiled by the National Cancer Institute (NCI). Additionally, the *in vitro* evaluation of **MS47** and **MS49** in the human melanoma MDA-MB-435 cell line using an MTT assay, clonogenic assay, cell cycle analysis, annexin V-FITC/PI apoptosis assay, and senescence-associated β-galactosidase-staining assay is described. Western blot analyses were performed to test the hypothesis that binding to the G4-DNA Hsp90 gene promoter results in the down-regulation of Hsp90 protein upon the treatment of the human melanoma MDA-MB-435S cell line with **MS47** and **MS49**.

## 2. Materials and Methods

### 2.1. Synthesis of MS47 and MS49

Synthesis of **MS47** and **MS49** was described previously by Saleh et al. [17]. Briefly, the synthesis process required three main steps. The first step was synthesis of the hydrophobic aromatic core, bis-(*p*-aminobenzyltriazolo)benzoquinone moiety, which included: (i) heating *p*-nitrobenzyl bromide with sodium azide in acetone under reflux for 24 h to

yield 82% of *p*-nitrobenzyl azide; (ii) adding the azide product to *p*-benzoquinone (in the ratio 2:1) via 1,3-dipolar cycloaddition reaction in ethyl acetate under reflux condition to afford bis-(*p*-nitrobenzyltriazolo)benzoquinone; and (iii) reduction of bis-triazole product via palladium-catalysed hydrogenation at room temperature in DMF to produce a good yield (92%) of bis-(*p*-aminobenzyltriazolo)benzoquinone. The second step was synthesis of the protonable side arms, dialkylaminoacid chlorides, which involved: (i) treating the aminoesters, methyl 3-(diethylamino)propanoate and methyl 3-(piperidin-1-yl)propanoate, with potassium hydroxide at room temperature to afford the corresponding amino acids, 3-(diethylamino)propanoic acid and 3-(piperidin-1-yl)propanoic acid; and (ii) the aminoacids were treated with oxalyl chloride and few drops of DMF under inert atmosphere in dry $CH_2Cl_2$ to produce the corresponding aminoacid chlorides, 3-(diethylamino)propanoyl chloride and 3-(piperidin-1-yl)propanoyl chloride. The third step involved coupling the bis-triazol hydrophobic core with the aminoacid chlorides' side chains in the presence of pyridine catalyst under inert atmosphere at room temperature in dry DMF to produce good yields of novel compounds, bis-{*p*-[3-(diethylamino)propanamido]benzyltriazolo}benzoquinone (64%) and bis-{*p*-[3-(piperidin-1-yl)propanamido]benzyltriazolo}benzoquinone (68%), which were converted to the corresponding hydrochloride salts **MS47** and **MS49** in yields of 57% and 67%, respectively, by treating them with hydrogen chloride in methanol.

### 2.2. Cell Culturing

The MDA-MB-435 human melanoma cell line was obtained from the American Type Culture Collection (ATCC) (Manassas, VA, USA) and kept in liquid nitrogen. Upon achieving 70–80% confluence, cells were passaged twice weekly, and modest passage numbers ($\leq$20) were utilized in all tests. The use of Lonza MycoAlert$^{TM}$ mycoplasma detection kits (in accordance with the manufacturer's instructions) ensured that cultures were free of bacterial infection (contamination). Sub-cultured cells were grown in RPMI-1640 media (Sigma-Aldrich, Gillingham, UK, CN: R8758) supplemented with 10% (*v/v*) heat-inactivated FBS (Gibco, CA, USA). Cells were kept at 37 °C and 100% humidity in a 5% $CO_2$ incubator (Shel Lab., Cornelius, OR, USA).

### 2.3. Measuring Cell Viability (MTT Assay)

The chosen method used in this study was originally described by Saleh et al. [17,26] and adapted from Mosmann [20]. MTT reagent (3-(4,5-dimethylthiazol-2-yl)-2,5-diphenyl tetrazolium bromide) has been used to determine the antiproliferative activities of both **MS47** and **MS49** on human melanoma MDA-MB-435 cell line. Cells were seeded at a density of $3 \times 10^3$ per well (180 μL per well) into 96-well plates (Nunc, Roskilde, Denmark) and allowed to adhere at 37 °C/5% $CO_2$ for 24 h. Fresh top stock solutions of the test compounds (10 mM in DMSO) were prepared, followed by the preparation of serial dilutions in RPMI-1640 media for addition to cancer cells. The control wells only received the vehicle (20 μL media per well). Final concentrations of the test compound in the wells were: 0.01, 0.05, 0.1, 0.5, 1, 5, 10, 50 and 100 μM. While the final DMSO concentration in the wells never surpassed 1%. Vehicle control experiments were conducted (0.0001–1% DMSO). Test plates were incubated for further 72 h at 37 °C and 5% $CO_2$. At the time of test compound addition ($T_0$) and after 72 h exposure, cell viability was determined by the addition of MTT reagent (2 mg/mL in PBS; 50 μL per well). To allow the reduction in MTT by live cells to insoluble crystals of dark purple formazan, test plates were incubated for 3 h. Then, the supernatant was removed from each well by means of aspirator, and DMSO (150 μL per well) was added to solubilize the cellular formazan. The absorbance was measured at 550 nm using a plate reader from Anthos Labtec systems. The intensity recorded (OD) is proportional to metabolic activity, which coincides with the number of live cells. By using Microsoft Excel 2010 software, $GI_{50}$ values (test compound concentrations that inhibit cell growth by 50%) were estimated. Results are presented as mean of three separate experiments (*n* = 8 per trial).

### 2.4. Clonogenic Assay

The clonogenic cell survival test examines the capacity of a single cell to withstand a brief exposure to test chemicals and maintain its proliferative potential to generate offspring colonies [27,28]. The assay was conducted according to the prior description [29]. A total of 400 (melanoma MDA-MB-435) cells/well were sown into 6-well plates with 1.0 mL of media and left to adhere for 24 h prior to treatment with the test agent. After 24 h of exposure to **MS47** and **MS49** ($1 \times GI_{50}$ and $2 \times GI_{50}$ values), medium containing **MS47** and **MS49** was aspirated, cells were gently washed twice with 1.0 mL medium, and 2.0 mL of fresh medium were added prior to incubation for ~10–14 days. When colonies of more than 50 cells were visible in the control wells, experiments were halted. The colonies were washed twice with 1.0 mL of cold PBS, fixed with 100% methanol (1.0 mL/well) for 10 min, stained with 0.5 mL of 0.05% methylene blue (1:1 d-$H_2O$:methanol; 15 min), washed ($3\times$ d$H_2O$), air dried and counted. Prism software was used to calculate the mean survival fraction (%) of untreated and treated cell colonies.

### 2.5. Western Blots

Western blotting was performed according to Chen et al.'s instructions [30]. In 10 cm$^2$ culture dishes, cells were sown at a density of $1$–$2 \times 10^6$ per dish, allowed to adhere for 24 h, and exposed to $1 \times GI_{50}$ and $2 \times GI_{50}$ of **MS47** and **MS49** for 72 h. After exposure to the treatment, protein lysates were prepared, and protein concentrations were determined using the Bradford assay [31]. Using SDS PAGE, 50 μg of protein per sample was separated, then transferred to a nitrocellulose membrane. The 1° antibodies (Abs) Hsp90 and GAPDH were bought from Cell Signalling Technology (Danvers, MA, USA). While Dako (Santa Clara, CA, USA) supplied the 2° antibody (Ab) horseradish peroxidase-conjugated anti-rabbit immunoglobulin G (IgG). Immunoblotting was used to detect proteins, as previously described by Mahmood et al. [32]. The densitometric analysis was performed by using the ImageJ software (NIH, Bethesda, MD, USA).

### 2.6. Cell Cycle Analysis by Flow Cytometry

Cell cycle analysis was performed using a fluorochrome solution comprising 50 mg/mL propidium iodide (PI), 0.1% (*v/v*) Triton X-100, 0.1 mg/mL of ribonuclease A and 0.1% (*w/v*) sodium citrate in d-$H_2O$ according to Nicoletti et al.'s technique [33]. A total of $1 \times 10^5$ cells/well were seeded in 6-well plates and incubated overnight before being treated with **MS47** and **MS49** at $0.5 \times$, $1 \times$ and $2 \times GI_{50}$ doses for 24, 48 and 72 h. At the end of the specified time period, cells were extracted, pelleted by centrifugation, then resuspended in 0.3–0.5 mL of fluorochrome solution, and kept in the dark at 4 °C overnight. Beckman Coulter FC500 flow cytometer was utilized for cell cycle analyses (Indianapolis, IN, USA). FCS express analysis software version 7 (De Novo software, Pasadena, CA, USA) was used to analyze the data, and model 1 of multivariate analysis was employed as the mathematical model. The analysis was performed automatically by selecting the best-fitting cycle, one cycle at a time, and displaying only the major phases of the cell cycle, G0/G1, S, and G2/M.

### 2.7. Annexin V-FITC and Propidium Iodide Apoptosis Assay

In 6-well plates, cells were sown at a density of $1.0 \times 10^5$ cells/well, and incubated overnight prior to treatment with **MS47** and **MS49** at $0.5 \times$, $1 \times$ and $2 \times GI_{50}$ doses, for 24, 48 and 72 h. After treatment, cells were trypsinized, centrifuged, collected in FACS tubes and maintained in 2 mL of cold media on ice for 10 min. After centrifugation, the cells were rinsed with ice-cold PBS and subsequently pelleted by centrifugation. Annexin-V-FITC (5 μL) and $1\times$ annexin-V buffer (100 μL) were applied to cells; after 15 min of incubation at room temperature in the dark, PI (10 μL; 50 μg/mL in PBS) and 400 μL annexin-V buffer were added. Prior to analysis with Beckman Coulter FC500 flow cytometer, cells were maintained on ice and in the dark for 10 min (Indianapolis, IN, USA). Using version 7 of FCS express analysis software, data were analyzed (De Novo software, USA).

*2.8. Senescence-Associated β-Galactosidase Staining*

Investigation of senescence-associated (SA) β-galactosidase (SA-β-Gal) followed the chromogenic experiment which was published by Debacq-Chainiaux et al. [34] using light microscopy. The cells were seeded at a density of $1 \times 10^4$ cells/well in 6-well plates, incubated overnight, and then treated with **MS47** and **MS49** at $0.5 \times$ and $1 \times$ GI$_{50}$ concentrations for 72 h. After 72 h, the medium containing treating ligands was aspirated, cells were washed twice with $1\times$ PBS (2 mL per well), $1\times$ fixation solution (containing 2% formaldehyde/0.2% glutaraldehyde in PBS) was added (2 mL per well), and then the cells were left to fix at room temperature for 10–15 min. The fixation solution was eliminated, cells were washed twice with $1\times$ PBS, then 1.0 mL of the freshly prepared staining solution (SA) β-galactosidase was added to each well, and cells were incubated at 37 °C for 12–16 h in a dry incubator (no CO$_2$). While the staining solution (SA) β-galactosidase was still on the plate, the cells were examined using a phase contrast microscope ($10\times$ magnification) for the appearance of a blue hue. Using Prism software or Microsoft Excel, the percentage of stained cells was determined.

*2.9. Statistical Analyses*

Experiments were performed a minimum of three times, with sample experiments depicted in the Figures. The statistical significance was determined using one-way and two-way analyses of variance (ANOVAs). Using Tukey's multiple comparison test, levels of significance (* $p < 0.05$, ** $p < 0.005$, *** $p < 0.0005$ and **** $p < 0.00005$ compared to untreated control) were identified. The significance level for clonogenic assay was determined using Dunnett's multiple comparison test (**** $p < 0.0001$ compared to untreated control).

**3. Results and Discussion**

*3.1. NCI 60 Cell Line Panel In Vitro Screening*

The NCI *in vitro* screen comprises a panel of sixty distinct human tumor cell lines (from nine different organ sites), against which compounds are tested over a specific concentration range (10 nM–100 μM) to evaluate the relative degree of growth inhibition or cytotoxicity against each cell line. For each studied drug, a characteristic profile or "fingerprint" of cellular response is established [35]. The "dose–response curves" and "mean graphs" are the components of the NCI screening data report package [36] that most investigators are interested in. The dose–response curve provides three response parameters: the 50% growth inhibition (GI$_{50}$), total growth inhibition (TGI), and 50% lethal concentration (LC$_{50}$) "net cell killing" or "cytotoxicity parameter", which are the concentrations of the test drug at which the percentages of the cell growth (PG) are +50%, 0%, and −50%, respectively [37]. The mean graph is a pattern made by plotting positive and negative values, referred to as "deltas", obtained from PG, GI$_{50}$, TGI or LC$_{50}$ data for a given chemical evaluated against each cell line in the NCI *in vitro* screen. Deltas are represented as horizontal bars in relation to a vertical line representing the mean panel [35]. Each bar represents the divergence of a single cancer cell line from the overall mean value of all the tested cells. The mean graph displays the relative resistance and sensitivity of all the cell lines at three levels of effect: GI$_{50}$, TGI and LC$_{50}$. Relatively sensitive cell lines have negative deltas and right-extending bars. While relatively resistant cell lines have positive deltas and bars that extend to the left [38].

The methodology of NCI 60 panel *in vitro* screening consists of two consecutive stages: the NCI 60 cell one-dose screen and the NCI 60 cell five-dose screen [35]. In the first phase, all the submitted compounds are initially evaluated at a single high concentration ($10^{-5}$ M) against a panel of sixty distinct human cancer cell lines. The data from a single dose are presented as a mean graph of the percent growth (PG) of the treated cells, allowing for the detection of both growth inhibition (values > 0) and fatality (values < 0). For instance, a score of 100 indicates no growth inhibition, 40 indicates 60% growth inhibition, 0 indicates no net growth throughout the period of the experiment, −40 indicates 40% lethality (cytotoxicity), and −100 indicates all cells are dead [39]. Only compounds that

meet the pre-determined inhibitory threshold requirements in a minimum number of cell lines will advance to the second phase. Compounds that display significant growth inhibition in the one-dose screen are routinely tested against the panel of sixty cells at five 10-fold dilutions (10 nM, 100 nM, 1 µM, 10 µM and 100 µM) in the five-dose screen. Each test compound in the comprehensive screen produces sixty dose-response curves and three mean graphs of the three response parameters $GI_{50}$, TGI and $LC_{50}$ [35,38,40].

Bis-triazole **MS47** was selected by the National Cancer Institute for testing against the NCI 60 panel of human tumor cell lines. The mean graph of the initial single-dose (10 µM) screen of **MS47** (named as NSC778438) is presented in Figure 2. The percentage growth (PG) that is altered due to treatment represents the anticancer activity. The majority of melanoma and CNS cancer cell lines are relatively very susceptible to **MS47**. The highest cytotoxic activity for **MS47** (yellow highlight) is 96.85% against the melanoma cell line SK-MEL-5, followed by 94.25% against the breast cancer BT-549 cell line, followed by 92.76% against the CNS cancer SNB-75 cell line and 92.63% against the prostate cancer DU-145 cell line. Although leukemic cell lines showed resistance to the lethality of **MS47** ($LC_{50}$ > 100 µM) (Table 1), **MS47** demonstrates the highest growth inhibition of 97.98% against the leukemia K-562 cell line of all the tested cells, followed by the breast cancer HS 578T cell line, and the renal cancer A498 and RXF 393 cell lines with growth inhibitions of 85.84%, 85.43% and 80.06%, respectively.

Renal cancer cell lines demonstrate an exceptional resistance to **MS47**. The renal cancer TK-10 and UO-31 cell lines, ovarian cancer NCI/ADR-RES cell line, and colon cancer HCT-15 cell line have the lowest growth inhibition and highest relative resistance.

Ligand **MS47** (NSC778438) satisfied the pre-determined criteria for the threshold inhibition in a minimum number of cells in the NCI 60 cell one-dose screening, so it was tested against the panel of sixty tumor cell lines of NCI at five small doses. Table 1 and Figure 3 illustrate the $GI_{50}$, TGI and $LC_{50}$ values (µM) and dose–response curves of the **MS47** (NSC778438) against the panel of NCI 60 human cancer cell lines that resulted from the NCI 60 cell five-dose screening. While the mean graphs of the $\log_{10}$ values (Molar) of $GI_{50}$, TGI and $LC_{50}$ of **MS47** (NSC778438) obtained from the NCI 60 cell line experiments can be found in the Supplementary Materials (Figure S1).

The results in Table 1 and dose–response curves in Figure 3 reflect the potent anti-cancer activity (cellular growth inhibition, total growth inhibition and lethality) of **MS47** (NSC778438) against most of the NCI 60 human cancer cell lines. The NCI 60 $GI_{50}$ values range from 0.111 µM to 24.1 µM with the exception of the ovarian cancer NCI/ADR-RES and renal cancer CAKI-1 cell lines, which showed significant resistance to **MS47** with $GI_{50}$ values > 100 µM. The most sensitive cell line to **MS47** was the melanoma MALME-3M cell line, with the lowest sub-micromolar $GI_{50}$ value being 0.111 µM, followed by the ovarian cancer OVCAR-4, breast cancer MCF7 and MDA-MB-468, colon cancer COLO 205, CNS cancer U251, SNB-75 and SNB-19, melanoma MDA-MB-435 and LOX IMVI, renal cancer SN12C, and non-small cell lung cancer NCI-H522 cell lines, showing sub-micromolar $GI_{50}$ values < 0.2 µM. The least growth inhibitory activity of **MS47** was for the renal cancer 786-0 cell line ($GI_{50}$ = 24.1 µM).

The total growth inhibition activity of **MS47** (NCS778438) against the NCI 60 human cancer cell lines can be described by the NCI 60 TGI values, which range from 0.26 µM to 38.1 µM. Resistance to **MS47** (TGI values > 100 µM) can be found in the leukemia CCRF-CEM, colon cancer HCT-15, ovarian cancer NCI/ADR-RES, and renal cancer 786-0 and CAKI-1 cell lines. **MS47** showed the highest potency in total growth inhibition against the ovarian cancer OVCAR-4 cell line (TGI = 0.26 µM), followed by the melanoma MALME-3M cell line (TGI = 0.265 µM), and other cell lines showing sub-micromolar TGI values less than 0.4 µM. While the lowest potency in the total growth inhibitory activity was against the renal cancer UO-31 cell line.

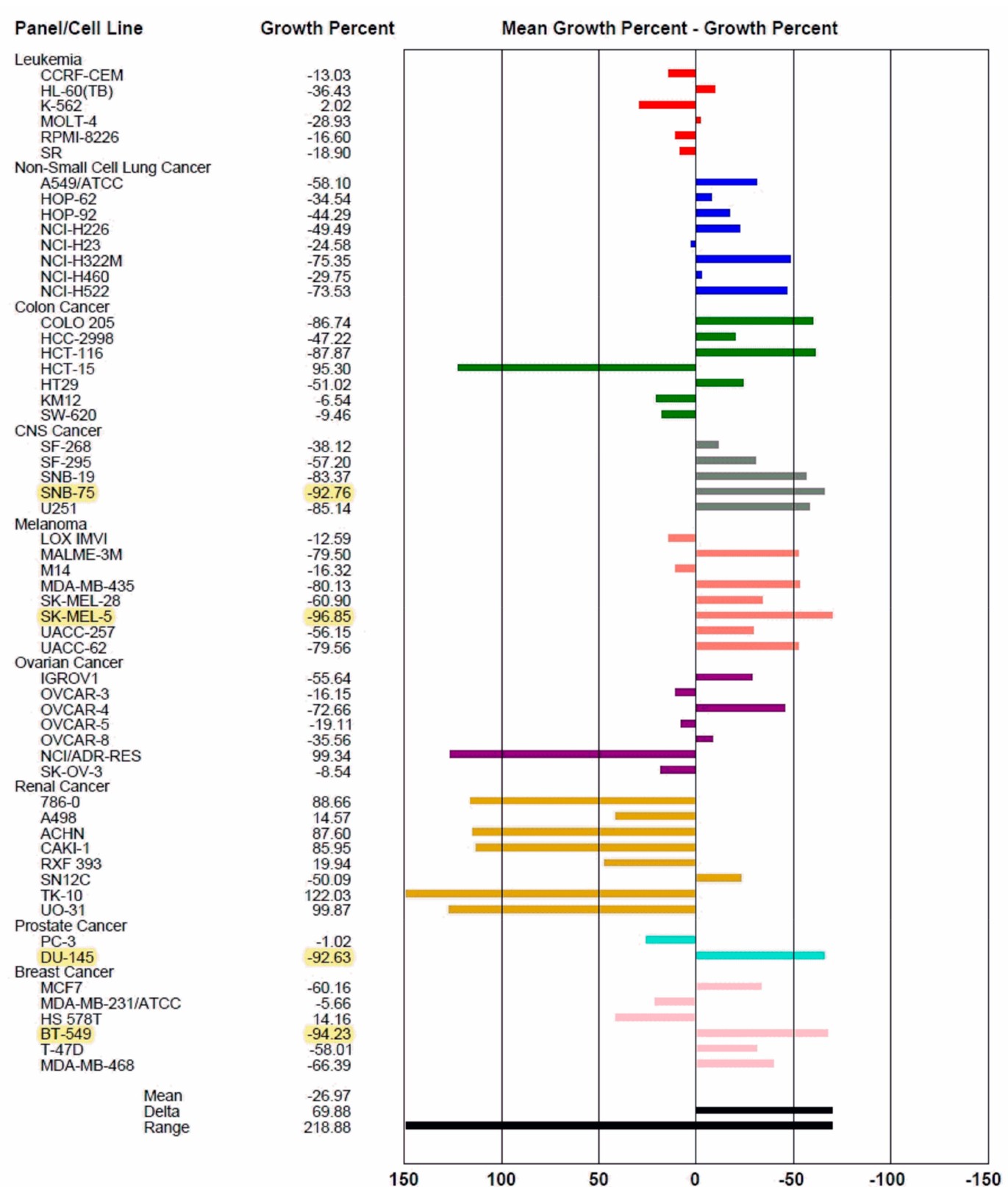

**Figure 2.** The mean graph of the one-dose (10 µM) screen of **MS47** (NSC778438) illustrates the sensitivity of the sixty human tumor cell lines to the cytotoxic activity of **MS47**. Yellow highlights represent the most sensitive cell lines to **MS47** cytotoxic (lethal) activity and their PG values in the range of 90%.

The NCI 60 $LC_{50}$ values show the lethality and cytotoxicity of **MS47** (NCS778438) against the NCI 60 human cancer cell lines. The values range from 0.515 µM to 86.1 µM. The ovarian cancer OVCAR-4 cell line showed the highest sensitivity to **MS47** ($LC_{50}$ = 0.515 µM), followed by the melanoma MDA-MB-435 and MALME-3M, colon cancer COLO 205, and CNS cancer U251 cell lines, with sub-micromolar $LC_{50}$ values of 0.602, 0.636, 0.644 and 0.694 µM, respectively, and other cell lines with $LC_{50}$ values of <1 µM. The least lethal activity shown by **MS47** was for the renal cancer UO-31 cell line ($LC_{50}$ = 86.1 µM). All of the leukemia cell lines showed interesting resistance to **MS47** cytotoxic activity, in addition to the colon cancer HCT-15, ovarian cancer NCI/ADR-RES, renal cancer 786-0 and CAKI-1, prostate cancer PC-3 and breast cancer HS 578T cell lines, with $LC_{50}$ values of >100 µM. The resistance and sensitivity shown by the panel of sixty human cancer cell lines reflected a significant selective cytotoxic (lethal) activity for **MS47**.

The NCI 60 cell five-dose screening results indicated that **MS47** (NSC778438) has potent and selective cell growth inhibition and cytotoxic activities against tumor cell lines isolated from distinct organs, making it a promising anticancer drug candidate for further development for the treatment of multiple carcinomas, such as renal, melanoma, ovarian, colon, breast, and CNS cancers.

The cytotoxicity of **MS47** indicated by the NCI 60 panel *in vitro* screen results and the higher selective potent growth inhibitory shown by **MS49** indicate that further preclinical evaluations of both ligands are warranted. Specifically notable is the sensitivity of the melanoma MDA-MB-435 cell line ($LC_{50}$ = 0.602 µM); thus, this cell line was selected for more thorough interrogation of anticancer activity.

**Table 1.** The anticancer activity of **MS47** (NSC778438) against the NCI 60 human cancer cell lines illustrated by its $GI_{50}$, TGI and $LC_{50}$ values (µM). Yellow highlights represent the highest potency of **MS47**, while green highlights represent its lowest potency against NCI 60 panel.

| Panel/Cell Line | Response Parameters of Ligand MS47 (µM) | | |
|:---:|:---:|:---:|:---:|
| | $GI_{50}$ | TGI | $LC_{50}$ |
| **Leukaemia** | | | |
| **CCRF-CEM** | 0.251 | >100 | >100 |
| **HL-60(TB)** | 0.226 | 0.643 | >100 |
| **K-562** | 0.597 | 14.4 | >100 |
| **MOLT-4** | 0.226 | 0.600 | >100 |
| **RPMI-8226** | 0.629 | 3.86 | >100 |
| **SR** | 0.457 | 5.35 | >100 |
| **Non-Small Cell Lung Cancer** | | | |
| **A549/ATCC** | 1.43 | 3.17 | 7.04 |
| **EKVX** | 1.54 | 3.05 | 6.04 |
| **HOP-62** | 1.54 | 2.91 | 5.52 |
| **HOP-92** | 0.423 | 1.40 | 5.88 |
| **NCI-H226** | 1.56 | 3.12 | 6.24 |
| **NCI-H23** | 0.267 | 0.794 | 24.8 |
| **NCI-H322M** | 0.289 | 1.04 | 3.45 |
| **NCI-H460** | 0.344 | 1.38 | 6.24 |
| **NCI-H522** | 0.194 | 0.481 | 4.76 |
| **Colon Cancer** | | | |

**Table 1.** *Cont.*

| Panel/Cell Line | Response Parameters of Ligand MS47 (μM) | | |
|---|---|---|---|
| | GI$_{50}$ | TGI | LC$_{50}$ |
| COLO 205 | 0.174 | 0.335 | 0.644 |
| HCC-2998 | 0.317 | 1.12 | 4.49 |
| HCT-116 | 0.283 | 1.14 | 3.57 |
| HCT-15 | 22.9 | >100 | >100 |
| HT29 | 0.321 | 0.900 | 33.8 |
| KM12 | 0.544 | 2.17 | 6.39 |
| SW-620 | 0.407 | 1.73 | 7.40 |
| CNS Cancer | | | |
| SF-268 | 0.326 | 1.45 | 6.88 |
| SF-295 | 1.36 | 2.72 | 5.46 |
| SF-539 | 0.202 | 0.420 | 0.875 |
| SNB-19 | 0.182 | 0.371 | 0.755 |
| SNB-75 | 0.176 | 0.421 | 1.01 |
| U251 | 0.175 | 0.348 | 0.694 |
| Melanoma | | | |
| LOX IMVI | 0.187 | 0.489 | 2.62 |
| MALME-3M | 0.111 | 0.265 | 0.636 |
| M14 | 1.58 | 3.38 | 7.24 |
| MDA-MB-435 | 0.176 | 0.326 | 0.602 |
| SK-MEL-2 | 0.219 | 0.553 | 8.81 |
| SK-MEL-28 | 0.913 | 2.46 | 6.24 |
| SK-MEL-5 | 0.227 | 0.705 | 2.70 |
| UACC-257 | 0.711 | 2.42 | 7.07 |
| UACC-62 | 1.05 | 2.56 | 6.26 |
| Ovarian Cancer | | | |
| IGROV1 | 0.233 | 0.687 | 3.79 |
| OVCAR-3 | 0.261 | 0.761 | 7.37 |
| OVCAR-4 | 0.131 | 0.260 | 0.515 |
| OVCAR-5 | 0.592 | 2.52 | 9.07 |
| OVCAR-8 | 1.56 | 3.66 | - |
| NCI/ADR-RES | >100 | >100 | >100 |
| SK-OV-3 | 1.99 | 5.32 | 36.9 |
| Renal Cancer | | | |
| 786-0 | 24.1 | >100 | >100 |
| A498 | 1.61 | 6.02 | 26.0 |
| ACHN | 9.56 | 23.2 | 54.6 |
| CAKI-1 | >100 | >100 | >100 |
| RXF 393 | 3.60 | 15.2 | 54.4 |
| SN12C | 0.185 | 0.371 | 0.745 |
| TK-10 | 11.1 | 30.1 | 81.3 |

**Table 1.** *Cont.*

| Panel/Cell Line | Response Parameters of Ligand MS47 (μM) | | |
|---|---|---|---|
| | GI$_{50}$ | TGI | LC$_{50}$ |
| **UO-31** | 16.9 | 38.1 | 86.1 |
| **Prostate Cancer** | | | |
| **PC-3** | 2.40 | 7.52 | >100 |
| **DU-145** | 1.03 | 2.23 | 4.83 |
| **Breast Cancer** | | | |
| **MCF7** | 0.144 | 0.358 | 0.894 |
| **MDA-MB-231/ATCC** | 0.253 | 10.2 | 61.4 |
| **HS 578T** | 2.88 | 9.17 | >100 |
| **BT-549** | 1.73 | 3.16 | 5.78 |
| **MDA-MB-468** | 0.173 | 0.382 | 0.843 |

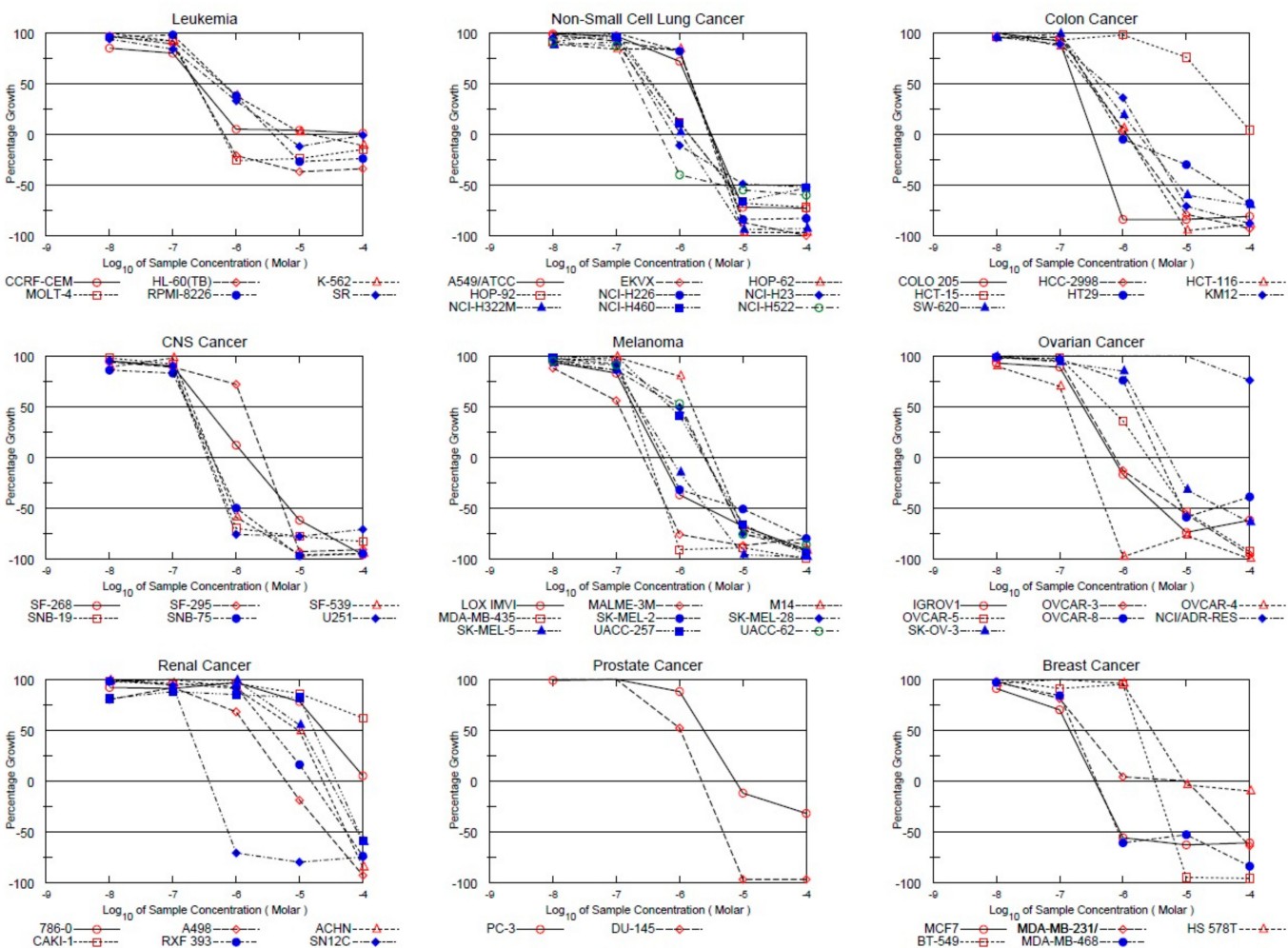

**Figure 3.** Dose–response curves of the cytotoxic activity of **MS47** (NSC778438) against the panel of sixty human cancer cell lines. Value of 100 for the growth percentage indicates the growth of untreated cells, while a growth percentage value of 0 indicates no net growth throughout the period of the experiment, and a growth percentage value of −100 represents that all of the cells were killed by **MS47**.

### 3.2. MTT Assay

The growth inhibitory activities of **MS47** and **MS49** were tested further *in vitro* using an MTT assay [20,41,42] against the human melanoma MDA-MB-435 cell line, which was chosen in particular for further *in vitro* anticancer evaluation of both ligands as it is one of the most sensitive cell lines to the lethal effect of **MS47**, as shown by the NCI 60 panel data.

The estimated concentrations at 50% cell growth inhibition ($GI_{50}$) were determined from the dose–response curves (Figure 4) after 72 h exposure of melanoma MDA-MB-435 cells to **MS47** and **MS49**, and are provided in Table 2. The $GI_{50}$ values resulting from examination of the growth inhibition by both ligands in human non-tumorigenic MRC-5 (embryonic lung fibroblasts) to evaluate their putative selective cytotoxic activities are also presented in Table 2. In the concentration range (0.01–100 µM), both **MS47** and **MS49** inhibit the growth of the human melanoma MDA-MB-435 cell line potently. However, ligand **MS49** ($GI_{50}$ value of 75 nM) shows more potent growth inhibitory effect than **MS47** ($GI_{50}$ value of 226 nM). Comparing the $GI_{50}$ values with those of human normal lung MRC-5 fibroblasts, both ligands demonstrate greater potency in the melanoma cell line than in non-tumorigenic lung fibroblasts, with indicated cancer selectivity indices (SI) of 9.8 for **MS47** and 17.7 for **MS49**, revealing their good selective growth inhibitory effects for cancer cells over normal cells.

**Table 2.** Growth inhibition effects of **MS47** and **MS49** on human melanoma MDA-MB-435 and human normal lung MRC-5 cell lines. Values of $GI_{50}$ are presented as mean ± standard deviation of at least three separate experiments (*n* = 8 per trial). "SI: Selectivity index ($GI_{50}$ MRC-5/$GI_{50}$ melanoma cell line)".

| Ligand | $GI_{50}$ (µM) ± S.D. | | SI |
| --- | --- | --- | --- |
| | **MDA-MB-435 Cell Line** | **MRC-5 Cell Line** | |
| **MS47** | 0.226 ± 0.057 | 2.219 ± 0.076 | 9.8 |
| **MS49** | 0.075 ± 0.010 | 1.325 ± 0.137 | 17.7 |

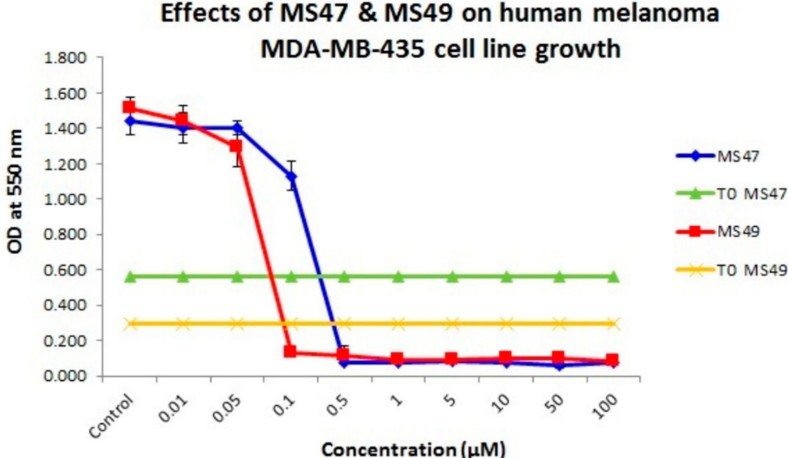

**Figure 4.** Dose–response curves that show the growth inhibiting effects of **MS47** and **MS49** against melanoma MDA-MB-435 cell line. Values are mean ± SD, *n* = 8, graphs are representative of experiments performed on at least three separate occasions.

### 3.3. Clonogenic Assay

The effects of **MS47** and **MS49** on cell survival and subsequent proliferative potential (progeny colony formation) were determined *in vitro* using the clonogenic cell survival assay [27,28]. **MS47** and **MS49** significantly inhibit MDA-MB-435 colony formation after 24 h exposure (Figure 5A,B). At values of 1 × $GI_{50}$ and 2 × $GI_{50}$, respectively, the colony

formation was similarly inhibited by both ligands (by 98.35% and 100%). This might indicate that for any circulating cells, brief exposure to a test agent may inhibit their ability to form metastatic tumor colonies. The two-way ANOVA found a significant effect of the ligands' concentration $F(2,22) = 87.77$, ($p < 0.0001$), a non-significant effect of the treatment $F(1,22) = 4.053 \times 10^{-30}$, ($p > 0.99$), and non-significant effect of interaction $F(2,22) = 1.53 \times 10^{-30}$, ($p > 0.999$). Dunnett's multiple comparison test determined the significance between control and $1 \times GI_{50}$ and $2 \times GI_{50}$ for both of the ligands. The clonogenic assay emphasized the anticancer cytotoxicity and potency of **MS47** and **MS49**, as the formation of colonies was significantly inhibited (**** $p < 0.0001$).

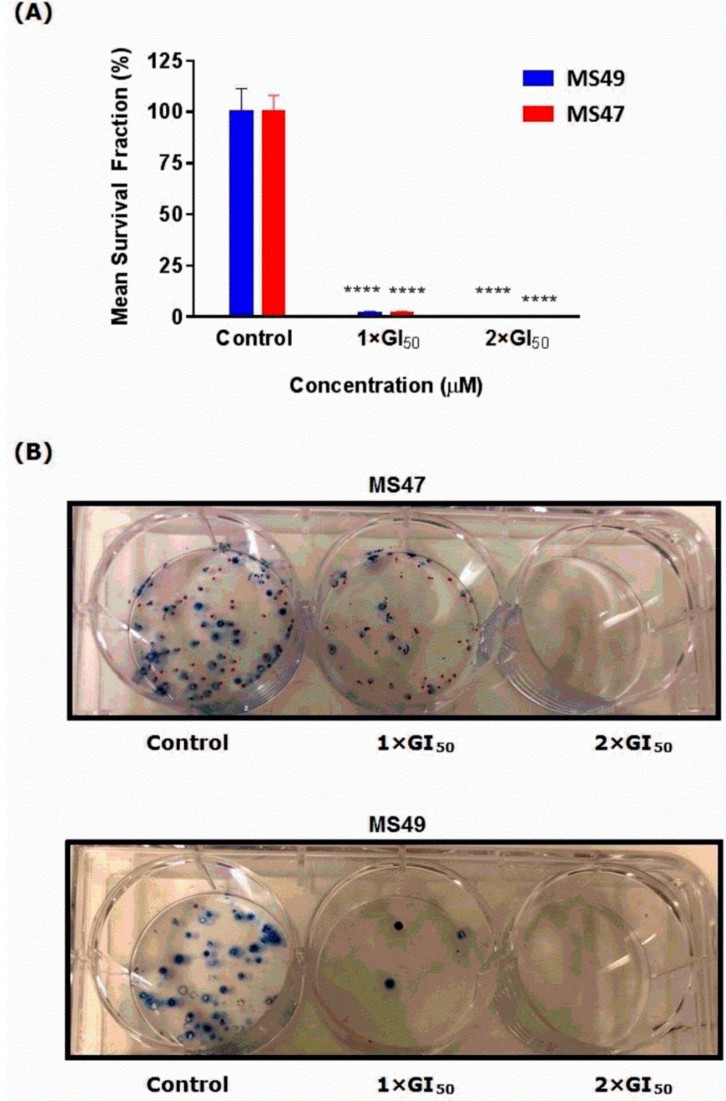

**Figure 5.** Effects of **MS47** and **MS49** on colony formation. (**A**) Mean $\pm$ SEM survival fraction (%) of the treated cells as a percentage of the control population for human melanoma MDA-MB-435 cells. **MS47** and **MS49** caused a significant inhibition in the colony formation (**** $p < 0.0001$, $n = 2$ for each of the three independent experiments). (**B**) Representative photographs show the effects of **MS47** and **MS49** on melanoma MDA-MB-435 colony formation in the control, $1 \times GI_{50}$ and $2 \times GI_{50}$.

### 3.4. Cell Cycle Analysis

To determine whether the MDA-MB-435 cell cycle was perturbed by treatment with **MS47** and **MS49** and help establish the cell death modality, both a cell cycle analysis and an apoptosis assay were performed [43,44].

The effect of treating the human melanoma MDA-MB-435 cells with **MS47** and **MS49** on the cell cycle was investigated by flow cytometry. Cells were exposed to **MS47** and **MS49** ($0.5 \times \mathrm{GI}_{50}$, $1 \times \mathrm{GI}_{50}$ and $2 \times \mathrm{GI}_{50}$) for 24, 48 and 72 h (Figure 6). Two-way ANOVA indicated that both **MS47** and **MS49** caused no significant change in the numbers of events in specific cell cycle phases among all treated groups compared to the untreated control group after 24 and 48 h of treatment (Figure 6A,B). In contrast, after 72 h of exposure, Tukey's multiple comparison test found that **MS47** ($2 \times \mathrm{GI}_{50}$) caused significant arrest in the G0/G1 and G2/M phases of the cell cycle in the human melanoma MDA-MB-435 cells (\*\* $p < 0.005$) and (\* $p < 0.05$), respectively (Figure 6C,D). While **MS49** ($2 \times \mathrm{GI}_{50}$) did not show any significant difference compared to the control untreated group and other treatments after 72 h of exposure. Thus, in terms of cell cycle arrest, **MS47** ($2 \times \mathrm{GI}_{50}$) exerts a more significant effect on treated cells in comparison to **MS49** ($2 \times \mathrm{GI}_{50}$) (\* $p < 0.05$). Consequently, we conclude that **MS47** may trigger cell death following the arrest of cells at G0/G1.

For both **MS47** ($2 \times \mathrm{GI}_{50}$) and **MS49** ($2 \times \mathrm{GI}_{50}$), we observed the presence of sub-G0/G1 events after 72 h, indicating the apoptotic death of the cancer cells. Additionally, the G2/M phase was depleted in both ligands in comparison to other treated groups in an observable manner. However, this depletion was significant only after treating the cells with **MS47** ($2 \times \mathrm{GI}_{50}$) (\* $p < 0.05$). Interestingly, the depletion of G2/M phase and the enhancement of the S, G0/G1, and sub-G0/G1 phases may indicate that our compounds triggered the apoptotic cell death. Cell cycle profiles are shown in the Supplementary Materials (Figure S2).

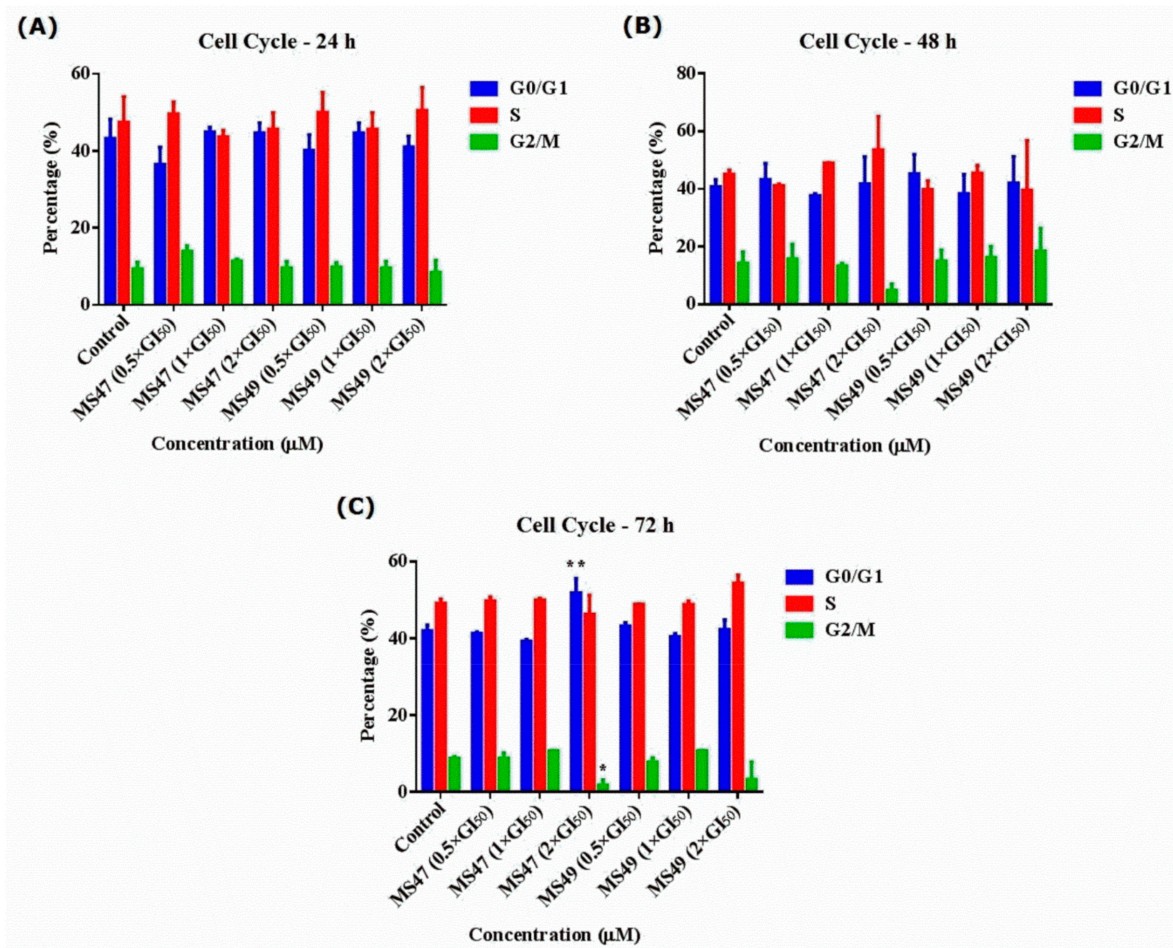

**Figure 6.** *Cont.*

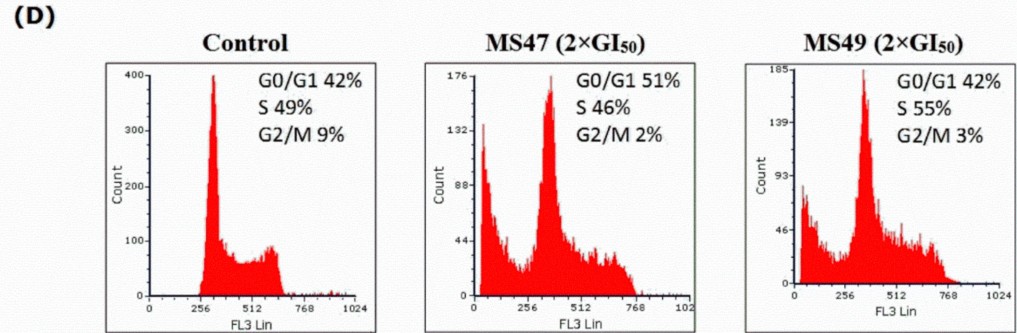

**Figure 6.** Statistical analyses of cell cycle phases: G0/G1, S and G2/M in human melanoma MDA-MB-435 cells, treated with $0.5 \times GI_{50}$, $1 \times GI_{50}$ and $2 \times GI_{50}$ of **MS47** and **MS49** compared to the untreated cells (control group) for (**A**) 24 h, (**B**) 48 h and (**C**) 72 h. **MS47** ($2 \times GI_{50}$) evoked significant arrest in the G0/G1 and G2/M phases (* $p < 0.05$ and ** $p < 0.005$; experiments were repeated $\geq$ three times, $n = 2$). (**D**) Flow cytometric cell cycle analysis (histograms) of human melanoma MDA-MB-435 cells, treated with **MS47** and **MS49** of $2 \times GI_{50}$ concentrations for 72 h compared to the control cells and stained with propidium iodide (PI).

*3.5. Annexin V-FITC and Propidium Iodide Apoptosis Assay*

To investigate the cell death mechanism and test the hypothesis that **MS47** and **MS49** may initiate apoptosis, human melanoma MDA-MB-435 cells were treated with both ligands ($0.5 \times GI_{50}$, $1 \times GI_{50}$ and $2 \times GI_{50}$) for 24, 48 and 72 h, stained with Annexin V-FITC/PI and analyzed by flow cytometry (Figure 7). Corresponding scatter plots are shown in the Supplementary Materials (Figure S3).

After 24 h of the treatment, the two-way ANOVA test found that **MS47** ($0.5 \times GI_{50}$) and ($2 \times GI_{50}$) have a direct effect on the cell viability as the number of healthy cells decreased significantly compared to the control untreated cells (* $p < 0.05$). Tukey's multiple comparison test shows that **MS47** ($0.5 \times GI_{50}$) and ($2 \times GI_{50}$) decreased the percentage of healthy cells compared to that recorded in the control untreated cells (* $p < 0.05$). Additionally, the percentage of late apoptotic cells was significantly increased in the treated group **MS47** ($2 \times GI_{50}$), as the number of apoptotic cells was ten-fold higher in the **MS47**-treated ($2 \times GI_{50}$) cells compared to the control cells (** $p < 0.005$). Whereas for **MS49**, the number of viable cells decreased significantly when the cells were treated with ($1 \times GI_{50}$) and ($2 \times GI_{50}$) **MS49** compared to the control untreated group at (* $p < 0.05$) and (** $p < 0.005$), respectively. Similarly to **MS47** ($2 \times GI_{50}$), **MS49** ($2 \times GI_{50}$) showed a 10-fold significant increase in the percentage of late apoptotic cells compared to that of the control untreated group (** $p < 0.005$), as shown in Figure 7A.

The same cytotoxic (apoptotic) effect was observed after 48 h of treatment. The two-way ANOVA found that **MS47** caused significant decrease in the percentage of viable cells (* $p < 0.05$), and a significant increase in the percentage of apoptotic cells compared to the control cells (** $p < 0.005$). Tukey's multiple comparison test found that the percentage of late apoptotic cells increased up to 77% (by two-fold) compared to the 24 h treatment (42%) (** $p < 0.005$) following exposure to **MS47** ($2 \times GI_{50}$). Similarly, **MS49** caused significant programmed cell death; the late apoptotic population increased, while the viable cell population decreased in a significant manner when the cells were treated with **MS49** ($2 \times GI_{50}$) for 48 h compared to the other treatments and the control untreated cells at (*** $p < 0.0005$) and (* $p < 0.05$), respectively (Figure 7B).

After 72 h exposure to **MS47** and **MS49**, the same cytotoxic (apoptotic) effect persisted. Tukey's multiple comparison test found that for the **MS47**-treated cells ($2 \times GI_{50}$), the percentage of viable cells decreased while the percentage of late apoptotic cells increased significantly compared to that of the control group (*** $p < 0.0005$). For **MS49**, $1 \times GI_{50}$ exposure caused a significant decrease in the viable cell population (* $p < 0.05$), accompanied by a significant increase the late apoptotic cells (*** $p < 0.0005$). **MS49** at $2 \times GI_{50}$ showed a

strong significant decrease in the number of viable cells compared to all the other treated groups (**** $p < 0.00005$), and a significant increase in the percentage of late apoptotic cells (*** $p < 0.0005$) (Figure 7C,D).

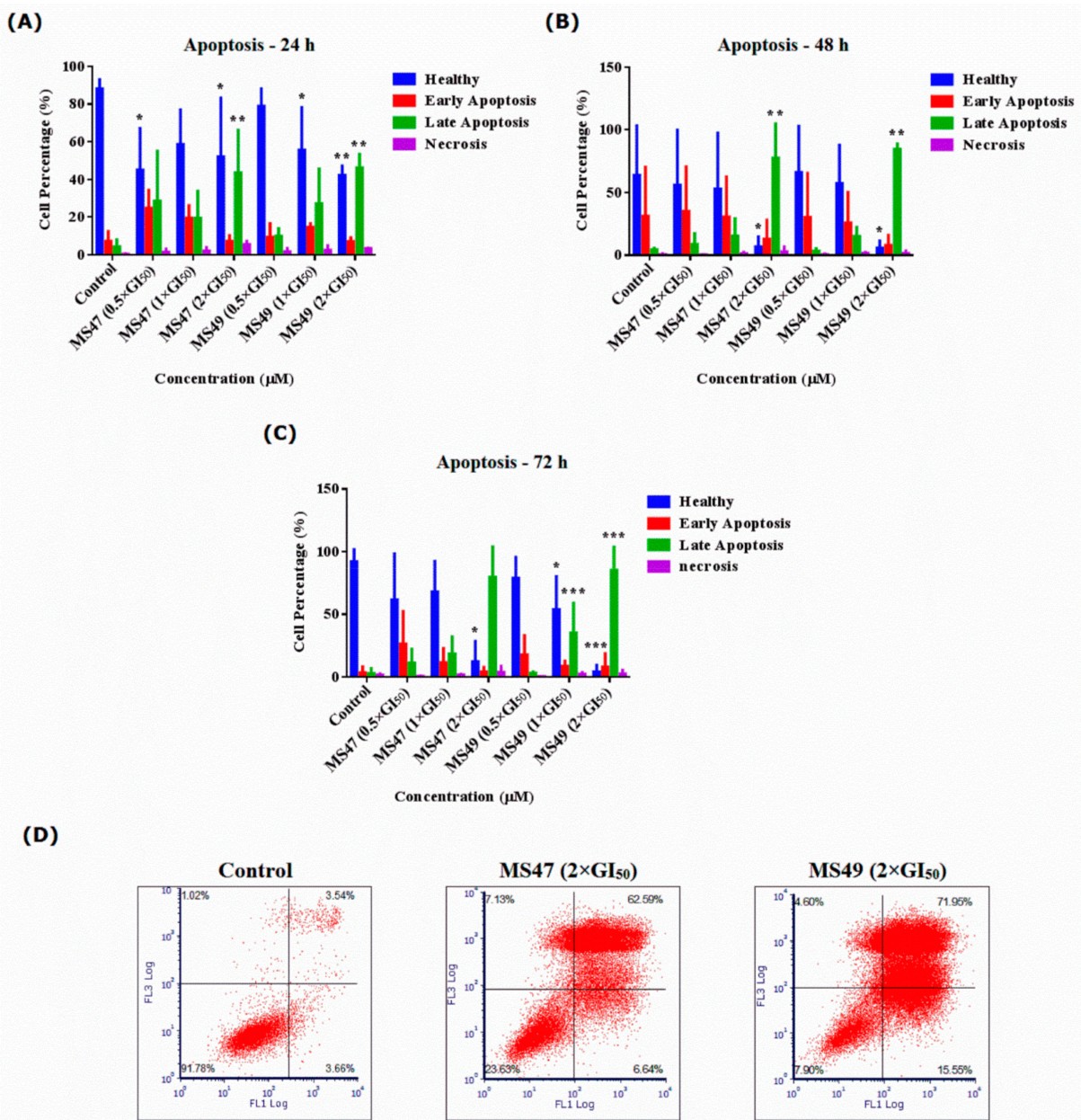

**Figure 7.** Statistical analysis of the percentages of healthy, apoptotic (early and late) and necrotic human melanoma MDA-MB435 cells, treated with **MS47** and **MS49** ($0.5 \times GI_{50}$, $1 \times GI_{50}$ and $2 \times GI_{50}$) compared to the untreated cells (control group) for (**A**) 24 h, (**B**) 48 h and (**C**) 72 h. Both ligands caused significant increase in the percentage and number of the late apoptotic cells (* $p < 0.05$, ** $p < 0.005$, *** $p < 0.0005$; experiments were repeated $\geq$ three times, $n = 2$). (**D**) Flow cytometric analysis (histograms) of cell apoptosis mechanism (apoptosis/necrosis) in human melanoma MDA-MB435 cells, treated with **MS47** and **MS49** of $2 \times GI_{50}$ concentrations compared to the untreated cells (control group) and stained with FITC-conjugated annexin V and propidium iodide (PI) for 72 h.

### 3.6. Western Blots

Heat-shock proteins (Hsps) are molecular chaperones that regulate protein folding to ensure correct conformation and translocation while preventing protein aggregation. Many solid tumors and haematological malignancies have an increase in heat-shock proteins [45]. Client proteins of Hsp90 include many oncogenic proteins involved in the cells' transformation into malignant forms. Thus, chemical inhibitors that target Hsp90, which may disrupt multiple oncogenic processes, would destroy these oncogenic proteins, making them effective as chemotherapeutic drugs [46].

Western blots [32] were performed to investigate the effect of our G-quadruplex stabilizers **MS47** and **MS49** on the expression levels of Hsp90 protein chaperone in human melanoma MDA-MB-435 cells. Protein lysates of melanoma cells following 72 h of exposure to **MS47** and **MS49** ($1 \times GI_{50}$ or $2 \times GI_{50}$ μM) were prepared. One-way ANOVA showed no significant effect of **MS47** treatment on the levels of Hsp90 protein in the human melanoma MDA-MB-435 cells (Figure 8A). However, the levels of the Hsp90 protein in melanoma cells were significantly decreased by $2 \times GI_{50}$ μM of **MS49**, compared to the cells incubated with the vehicle (control) $F_{(2,6)} = 5.841$, $p = 0.0391$ (Figure 8B). This significant decrease in the levels of Hsp90 protein can be explained by the ability of **MS49** (with piperidinyl group) to induce stabilization of the G-quadruplex formed by the Hsp90a promoter, with a binding affinity higher than that of **MS47** (with diethylamino group), as reported previously [17].

Blot images are splices to present the intended groups. The original image files for the blots are found in the Supplementary Materials (Figure S4).

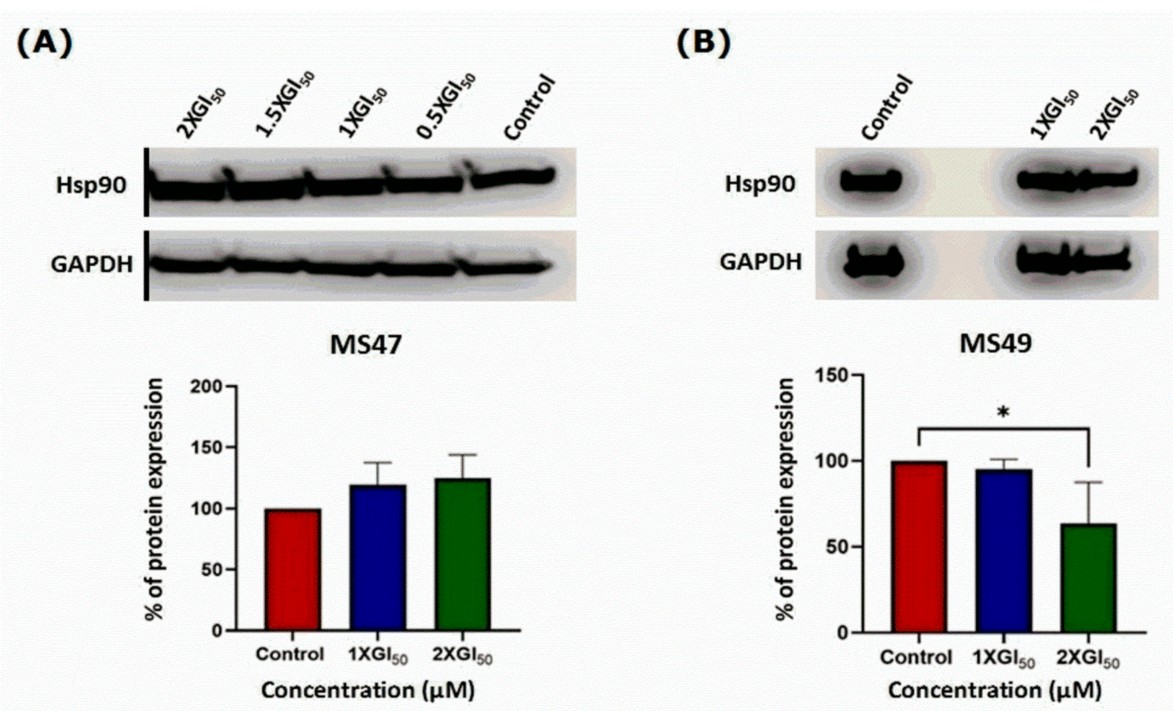

**Figure 8.** Hsp90 protein expression in human melanoma MDA-MB-435 cells; vehicle control and treated with ($1 \times GI_{50}$ and $2 \times GI_{50}$) of (**A**) ligand **MS47** and (**B**) ligand **MS49** for 72 h after incubation. Data are means $\pm$ SEM of three separate experiments (ANOVA statistical analysis followed by Tukey's test; * $p < 0.05$). Blot images are splices to present the intended groups. The original image files for the blots are found in the Supplementary Materials (Figure S4).

### 3.7. Senescence-Associated β-Galactosidase Staining

Due to cellular replication, telomeres shorten, resulting in cell cycle arrest (growth arrest), also known as replicative senescence [47]. Telomere length is stabilized and replenished by telomerase enzyme activity [48]. It has been revealed that G-quadruplex

stabilizers cause senescence in cancer cells by suppressing telomerase activity and speeding up telomere shortening, making them potentially valuable anticancer therapeutic agents [49]. Senescence-associated β-galactosidase (SA-β-Gal) staining [34] was performed to determine if our telomeric G4-DNA stabilizers **MS47** and **MS49** induce senescence in the human melanoma MDA-MB-435 cells. Figure 9 shows that neither **MS47** nor **MS49** at the concentrations tested ($0.5 \times GI_{50}$ and $1 \times GI_{50}$) induced senescence in the treated cells after 72 h of exposure as visualized by the absence of blue color. No statistical analyses were performed since the stained cells did not show any color. We speculate that the weak binding of both of the ligands to G4-DNA of the telomeric region may not be strong enough to stabilize the G-quadruplex; alternatively, the exposure period may not be long enough to impact the length of telomeres sufficiently to trigger senescence.

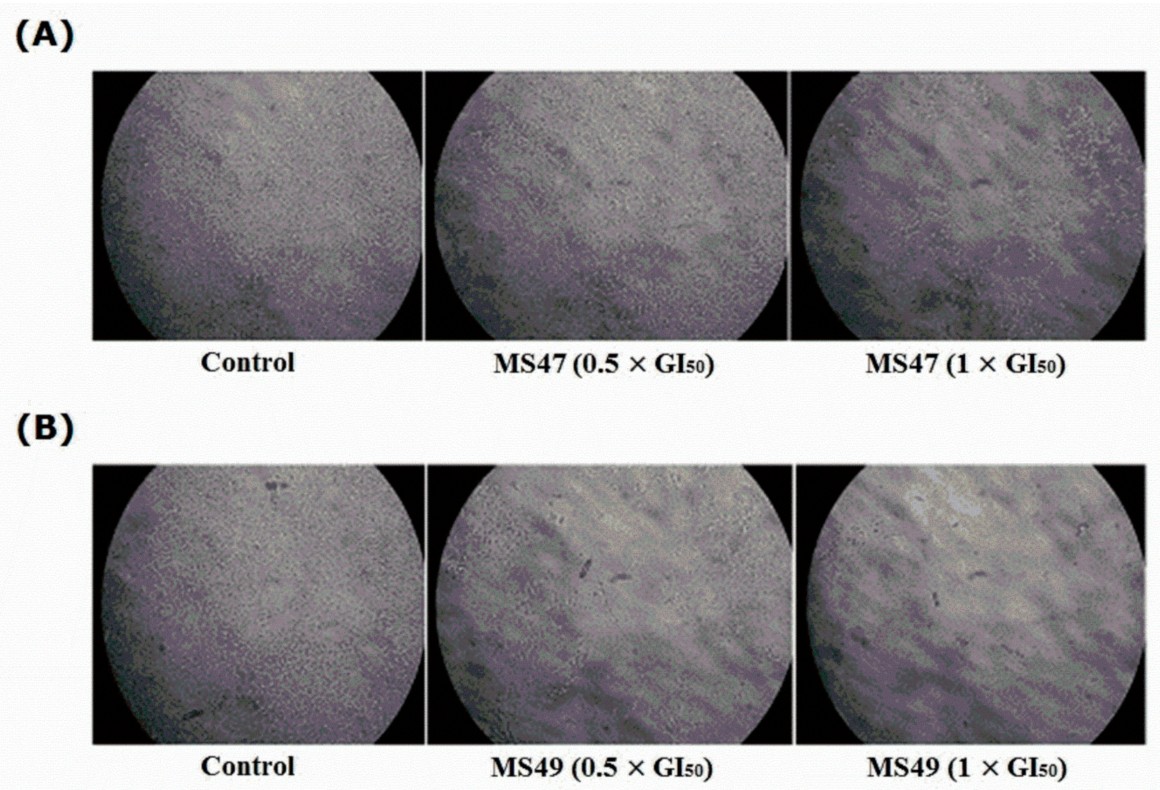

**Figure 9.** Human melanoma MDA-MB435 cells; untreated (control) and treated with ($0.5 \times GI_{50}$ and $1 \times GI_{50}$) of (**A**) ligand **MS47** and (**B**) ligand **MS49** for 72 h after incubation with β-galactosidase (SA) staining solution at 37 °C overnight (12–16 h) in a dry incubator (without $CO_2$). Treated cells did not show any blue color, as both ligands did not induce senescence.

## 4. Conclusions

Based on the observed activity of **MS47** (NSC778438), we demonstrate a promising novel compound with wide-spectrum selective cytotoxic activity against different types of cancer. Several cell lines showed sensitivity toward our drug. The NCI 60 $GI_{50}$, TGI and $LC_{50}$ values of **MS47** range from 0.111 μM to 24.1 μM, 0.26 μM to 38.1 μM and 0.515 μM to 86.1 μM, respectively. **MS47** displays the most selective lethal activity towards the ovarian cancer OVCAR-4 ($LC_{50} = 0.515$ μM) cell line, followed by the melanoma MDA-MB-435 and MALME-3M, colon cancer COLO 205, and CNS cancer U251 cell lines, with sub-micromolar $LC_{50}$ values of 0.602, 0.636, 0.644 and 0.694 μM, respectively, and other cell lines with $LC_{50}$ values less than 1 μM. In MTT assays, **MS47** and **MS49** demonstrated a potent cell growth inhibitory effect on the human melanoma MDA-MB-435, revealing selectivity (enhanced potency) over embryonic lung fibroblasts. Moreover, at $1 \times GI_{50}$ and $2 \times GI_{50}$ values, both ligands inhibited clonogenic cell survival (at $1 \times GI_{50}$ and $2 \times GI_{50}$

values) and induced significant apoptotic cell death. Western blot analyses showed that the G4 stabilizer **MS49** (at $2 \times GI_{50}$) significantly decreased the levels of Hsp90 in human melanoma MDA-MB-435 cells. **MS47** and **MS49** represent potent experimental antitumor agents, and can be used to treat cancer by inducing apoptotic cell death. Additionally, high concentrations of both ligands can be used as potent inhibitors for the progression of the cell cycle, as they have a role in the perturbation of the cell cycle after 72 h of treatment. To our knowledge, this is the first report on our prepared compounds **MS47** and **MS49**, which are novel and have not been synthesized by others. For our future work, the same screening will be repeated on other human melanoma cell lines, such as LOX IMVI and MALME-3M, to confirm if the observations/mechanisms of action are the same for multiple human melanoma cell lines. Furthermore, the molecular targets and mechanisms of action of anticancer bis-triazoles **MS47** and **MS49** in human melanoma MDA-MB-435 and other cancer cell lines will be interrogated. Our *in vitro* evaluation will include RNA sequencing (RNA-Seq) technology which quantitatively analyzes the whole-transcriptome sequencing (25,000 genes) of the RNA extracts of the treated cells to study the gene expression of cancer drug targets pertinent to tumorigenesis.

**Supplementary Materials:** The following are available online at https://www.mdpi.com/article/10.3390/cimb45010014/s1, Figure S1: Mean Graphs of $\log_{10}(GI_{50})$, $\log_{10}(TGI)$ and $\log_{10}(LC_{50})$ values (Molar) of **MS47** (NSC778438) obtained from the experiments of NCI 60 cell line. Figure S2: Flow cytometric cell cycle analysis of human melanoma MDA-MB-435 cells, treated with **MS47** and **MS49** of $0.5 \times GI_{50}$, $1 \times GI_{50}$ and $2 \times GI_{50}$ concentrations for (**A**) 24 h (**B**) 48 h and (**C**) 72 h compared to the control cells and stained with propidium iodide (PI). Figure S3: Flow cytometric analysis of cell death mechanism (Apoptosis/Necrosis) in human melanoma MDA-MB435 cells, treated with **MS47** and **MS49** of $0.5 \times GI_{50}$, $1 \times GI_{50}$ and $2 \times GI_{50}$ concentrations compared to the untreated cells (control group) and stained with FITC-conjugated annexin V and propidium iodide (PI) for (**A**) 24 h, (**B**) 48 h and (**C**) 72 h. Figure S4: The entire original gels for the western blot data. (**A**) Expression of Hsp90 after treatment with **MS47**. (**B**) Expression of Hsp90 after treatment with **MS49**.

**Author Contributions:** Performing research, M.M.S.; conceptualization, M.M.S. and T.D.B.; data curation, M.M.S. and M.S.H.; formal analysis, M.M.S., D.A.A., A.M.H., M.S.H., N.A. (Najneen Ahmed) and N.A. (Nazneen Aslam); funding acquisition, M.M.S. and C.A.L.; investigation, M.M.S.; methodology, M.M.S. and T.D.B.; project administration, M.M.S. and T.D.B.; resources, M.S.H., C.A.L. and T.D.B.; software, M.M.S., D.A.A., A.M.H., M.S.H., N.A. (Najneen Ahmed) and C.A.L.; supervision, C.J.M., C.A.L. and T.D.B.; validation, M.M.S.; visualization, M.M.S. and T.D.B.; writing—original draft preparation, M.M.S., D.A.A., A.M.H., A.Y.N., N.A. (Najneen Ahmed) and N.A. (Nazneen Aslam); writing—review and editing, A.Y.N., T.D.B., M.M.S., D.A.A. and A.M.H. All authors have read and agreed to the published version of the manuscript.

**Funding:** This research was sponsored and financially supported by research funding from University of Nottingham, Nottingham, UK, and Isra University, Amman, Jordan, to Maysaa M. Saleh. The project was conducted under Isra University President's Decision No. 27/1378/2016-2017; date of Grant: 20 July 2017.

**Institutional Review Board Statement:** Not applicable.

**Informed Consent Statement:** Not applicable.

**Data Availability Statement:** The data presented in this study are available in the article and Supplementary Materials.

**Acknowledgments:** The authors would like to thank the NCI Developmental Therapeutics Program (DTP), Washington, USA, for performing the 60 cell line screening, the University of Nottingham (UoN), Nottingham, UK, and Isra University, Amman, Jordan for access to their facilities and financial support.

**Conflicts of Interest:** The authors declare no conflict of interest.

## Abbreviations

| | |
|---|---|
| ANOVA | Analysis of variance |
| FITC | Fluorescein isothiocyanate |
| $GI_{50}$ | 50% Growth inhibition |
| HCT-116 | Human colorectal carcinoma cell line |
| Hsp90 | Heat shock protein 90 |
| Hsp90a | Labelled G-quadruplex forming oligomer of the promoter sequence Hsp90a oncogene |
| $LC_{50}$ | 50% Lethal concentration |
| MDA-MB-435 | Human melanoma cell line |
| MiaPaCa-2 | Human pancreatic carcinoma cell line |
| MRC-5 | Medical Research Council cell strain 5; embryonic lung fibroblast cell line |
| MTT | 3-(4,5-Dimethylthiazol-2-yl)-2,5-diphenyltetrazolium bromide |
| NCI | National Cancer Institute |
| OD | Optical density |
| PG | Percentage growth |
| PI | Propidium iodide |
| TGI | Total growth inhibition |

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
