# Peer review of "In Vitro Anticancer Properties of Novel Bis-Triazoles"

_cimb, doi:10.3390/cimb45010014_

Round 1

Reviewer 1 Report

The work by Saleh et al. is a large screening of the activity of two bis-triazoles. The detected cytotoxic and growth inhibitory activity against melanoma cell lines in the low micromolar or high nanomolar range.

Major points.

1. No normal cell line controls were used in the study, and thus it is hard to say anything about the selectivity and safety of the proposed compounds

2. The compounds induced up to 150% growth stimulation in the renal and some ovarian cancer cell lines. This behavior was not discussed in the text, and it raises major concerns about the clinical significance of the obtained results, i.e. the substances could stimulate the growth of cancer cells in the distant sites while killing the core tumor.

Minor points

1. The curves on the figure 3 are barely readable, please increase the resolution

2. The compound synthesis should be described in more detail, especially in the area of the reaction yields

Author Response

Dear Reviewer 1

Thank you very much your valuable comments and suggestions.

Kindly Regards

Dr. Maysaa Saleh

Reviewer 2 Report

Overall, the paper was easy to follow and provided some in vitro MOA of chemicals from a novel class of cancer therapeutics. 

Minior changes: 

-Explain why some cell lines are highlighted in yellow in Fig 2

 -Explain why MS49 was not tested in the NCI 60 panel; or have it tested as well

 -417/418 There is a repeating sentence

 -The conclusion statement could use some more summary of the data to tie everything together

Major changes: 

I recommend some of the screening be repeated on another human melanoma cell line like LOX IMVI which can be purchased to confirm if the observations/mechanisms of action are the same for multiple human melanoma cell lines. 

Author Response

Dear Reviewer 2

Many thanks for your valuable comments and suggestions.

Dr. Maysaa Saleh

Reviewer 3 Report

Some points must be revised:

- at lines 131-135, authors wrote: "Herein, we report the anticancer activity of MS47... described" Is this the aim of this paper? It is very difficult to understand what authors want to research in this paper. Revise.

- at line 202-206: "table 1 and Figure 3 reflect the potent anticancer activity...  "  Discuss more in the discussion section, what 's new in this paper?

- at line 223-234: "The NCI 60 LC50 values show the lethality and cytotoxicity of MS47 (NCS778438) against the NCI 60 human cancer cell lines.." Discuss more this part with other microenvironmental and cell tumor. Look at these refs: ---10.1016/S1607-551X(09)70106-8  --- doi: 10.3390/genes13061054 --- doi: 10.1016/j.peptides.2008.07.020

- at lines 546-548: " Additionally, high concentrations of both ligands can be... perturbation of the cell cycle after 72 h of treatmen" What does this paper add new to the literature? improve this part. What's new?

Author Response

Dear Reviewer 3

Many thanks for your valuable comments and suggestions.

Dr. Maysaa Saleh

Round 2

Reviewer 2 Report

Revised version is acceptable.

Reviewer 3 Report

ok